# Antibiotic Therapy for Active Crohn’s Disease Targeting Pathogens: An Overview and Update

**DOI:** 10.3390/antibiotics13020151

**Published:** 2024-02-03

**Authors:** Gaetano Iaquinto, Giuseppe Mazzarella, Carmine Sellitto, Angela Lucariello, Raffaele Melina, Salvatore Iaquinto, Antonio De Luca, Vera Rotondi Aufiero

**Affiliations:** 1Gastroenterology Unit, St. Rita Hospital, 83042 Atripalda, Italy; iaquintog@yahoo.it; 2Institute of Food Sciences, Consiglio Nazionale Delle Ricerche (CNR), 83100 Atripalda, Italy; gmazzarella@isa.cnr.it (G.M.); vera.rotondiaufiero@isa.cnr.it (V.R.A.); 3E.L.F.I.D, Department of Translational Medical Science, University “Federico II”, 80147 Napoli, Italy; 4Section of Human Anatomy, Department of Mental and Physical Health and Preventive Medicine, University of Campania “Luigi Vanvitelli”, 80138 Naples, Italy; antonio.deluca@unicampania.it; 5Department of Medicine, Surgery and Dentistry “Scuola Medica Salernitana”, University of Salerno, 84081 Salerno, Italy; 6Department of Sport Sciences and Wellness, University of Naples “Parthenope”, 80100 Naples, Italy; angela.lucariello@uniparthenope.it; 7Gastroenterology Unit, San G. Moscati Hospital, 83100 Atripalda, Italy; raffaelemelina@icloud.com; 8Gastroenterology Unit, St. Filippo Neri Hospital, 00135 Rome, Italy; salvatoreiaquinto@gmail.com

**Keywords:** Crohn’s disease, *Escherichia coli*, *Mycobacterium avium paratuberculosis*, antibiotic therapy

## Abstract

Crohn’s disease (CD) is a multifactorial chronic disorder that involves a combination of factors, including genetics, immune response, and gut microbiota. Therapy includes salicylates, immunosuppressive agents, corticosteroids, and biologic drugs. International guidelines do not recommend the use of antibiotics for CD patients, except in the case of septic complications. Increasing evidence of the involvement of gut bacteria in this chronic disease supports the rationale for using antibiotics as the primary treatment for active CD. In recent decades, several pathogens have been reported to be involved in the development of CD, but only *Escherichia coli* (*E. coli*) and *Mycobacterium avium paratubercolosis* (MAP) have aroused interest due to their strong association with CD pathogenesis. Several meta-analyses have been published concerning antibiotic treatment for CD patients, but randomized trials testing antibiotic treatment against *E. coli* and MAP have not shown prolonged benefits and have generated conflicting results; several questions are still unresolved regarding trial design, antibiotic dosing, the formulation used, the treatment course, and the outcome measures. In this paper, we provide an overview and update of the trials testing antibiotic treatment for active CD patients, taking into account the role of pathogens, the mechanisms by which different antibiotics act on harmful pathogens, and antibiotic resistance. Finally, we also present new lines of study for the future regarding the use of antibiotics to treat patients with active CD.

## 1. Introduction

Current data suggest that Crohn’s disease (CD) results from dysregulation of the mucosal immune system in genetically predisposed individuals, leading to strong and ongoing activation of the immunological response to intestinal microflora [1]. 

What triggers the onset of CD is still an open question, despite the progress that has been made in defining the genetic and environmental risk factors and understanding the pathways linked to the immune response regarding the inflammation aspect of the pathology. Several pathways are proposed to drive the disease [2]. 

The overall inflammatory response in CD could be an additional risk factor responsible for the development of the disease. In this regard, specific molecular events that regulate the production of cytokines, such as the loss of function mutations in the genes encoding interleukin (IL)-10 and its receptor (IL-10R), can cause early onset of CD. In addition, the regressive inheritance of rare and low-frequency deleterious NOD2 variants contributes to 7–10% of CD cases [3]. 

The inflammatory response in CD is due to the balance between key pro- and anti-inflammatory cytokines: tumor necrosis factor alpha (TNFα), IFN-γ, interleukin (IL)-1, IL-18, IL-33, IL-36, and IL-38, which have pro-inflammatory effects, and IL-10, IL-4, IL-6, IL-11, IL-13, and transforming growth factor beta (TGF-β), which have anti-inflammatory effects [3].

The cardinal symptoms of CD are severe abdominal pain, diarrhea, bleeding, bowel obstruction, and a variety of systemic symptoms affecting the mouth, eyes, joints, and skin. For decades, aminosalicylates, immunosuppressive agents, and corticosteroids have been the standard of care for active CD to control inflammation and induce clinical remission. The biological drugs that target cytokines, such as anti-TNFα, JAK inhibitors, monoclonal α4β7 integrin antibody, and anti-IL-12/IL-23, are part of the armamentarium to obtain clinical and endoscopic remission.

Regarding therapy for CD, the route of administration, how to choose the first and second biologics, the potential of combination therapy with biologics, and the safety of biologics have been recently reported in several articles [4,5,6]. However, the use of anti-TNFα therapy has not yielded the expected declines in hospitalization and intestinal resection in IBD [7].

In the last decades, several pathogens (Table 1) have been found to have a role in the pathogenesis of CD [8,9], but only *E. coli* [10,11,12,13] and *Mycobacterium avium paratubercolosis* (MAP) [14,15] have aroused interest due to their strong association with CD pathogenesis. In 1998, a new pathovar strain of *E. coli*, defined as adherent invasive *E. coli* (AIEC), was isolated from the ileal mucosa of CD patients, as that was assumed to be a potential etiological source of the disease [16]. AIEC was found to adhere to gut epithelial cells, invade mucosa, penetrate and replicate into macrophages, and release inflammatory cytokines [13,17,18,19]. 

It has been demonstrated that invasive *E. coli* strains isolated from CD patients are able to survive and replicate in large vacuoles within macrophages without inducing cell death. To survive and replicate in the harsh environment inside this compartment, AIEC strains utilize several adaptation mechanisms that permit them to resist phagocytosis and persist within macrophages, releasing large amounts of TNF-α [20].

Several independent studies, using different methods, reported an increased presence (from 25% to 55%) of mucosa-associated AIEC in CD patients [21,22,23]. AIEC was also recovered from 65% of chronic lesions and nearly 100% of biopsies from early lesions of CD patients [16]. In two recent reviews, AIEC was found in 23% and 29% of colonic mucosa biopsies from 69 and 304 CD patients, respectively [2,24]. All of these studies support the growing evidence that AIEC may be strongly involved in CD pathogenesis. Until now, few studies have been performed related to antibiotic treatment for active CD patients targeting AIEC. Unfortunately, the overall results are still scarce and unimpressive [25,26].

In addition to the presence of AIEC, several studies [27,28,29] reported the presence of MAP in intestinal biopsies of active CD patients, and for many years, it was also supposed that there may be an association between MAP and CD. Mycobacteria, like AIEC, survive and persist within host macrophages, and effective anti-mycobacterial agents require intracellular penetration. 

Recently, Khan et al. [2], using the RT-PCR method, found a significantly increased prevalence of MAP (23.2%) in biopsy samples from CD patients compared with non-IBD controls. Mycobacterial tuberculosis and MAP show different antibiotic sensitivities [30]. Several anti-MAP trials have been performed, some using a single drug and others using up to four drugs [31]. Although some trials and several case reports described mucosal healing and eradication of MAP [32], randomized trials with anti-MAP antibiotic treatment did not show any prolonged benefit for CD patients [33,34,35,36].

Townsend et al. showed that the outcome of short-term antibiotic treatment, which is useful for induction and remission of active CD, was uncertain [37]. Long-term antibiotic treatment trials have been also performed, but several questions were raised about the factors that could limit the effectiveness of antibiotic treatment: trial design, duration of treatment, dose, and combination of antibiotics. Until now, the choice of antibiotic treatment has always been arbitrary, and the primary endpoint was clinical and endoscopic remission.

In this paper, we provide an overview and update of the data from trials on antibiotic treatment of active CD, taking into account the role of pathogens in the progression of the disease and the mechanism of action of different antibiotics on harmful pathogens. This review takes a brief look at the past, present, and future of antibiotic-based therapies for patients with active CD.

Since we cannot exclude that the etiopathogenesis of CD may involve AIEC in some cases and MAP in others, we suggest that the choice of antibiotic treatment for active CD needs to consider the target pathogens. In fact, if the cause of the pathology is the presence of a specific bacterial species, eradication of that species would necessarily be beneficial for the regression of inflammation.

In the end, we tried to present new lines of study for the use of antibiotics with personalized therapy for CD patients, taking into account the presence or absence of a specific bacterial species.

## 2. Literature Search Strategy

A literature search was conducted using the National Institute of Health (NIH) website (http://www.clinicaltrials.gov, accessed on 8 December 2023) focused on antibiotic treatment targeting MAP and AIEC as an intervention in human trials with CD patients. There were no restrictions regarding language, research location, and research race. We carried out the bibliographic search from 2002 to 2023. 

The NIH database was chosen because it registers clinical trials around the world and the information is updated daily, and all of them are reviewed and approved by ethics committees or appropriate agencies and obey the appropriate national/state health agency regulations. We used an advanced search without any language restriction. The term “antibiotic Crohn” was entered into the search box. Studies that had no relation to antibiotic treatment were excluded. 

## 3. Antibiotic Treatment Targeting MAP in Active CD Patients

Several meta-analyses have been published concerning long-term antibiotic treatment targeting MAP in patients with active CD (Table 2). 

Borgaonkar et al. [33] identified six randomized controlled trials (RCTs) using anti-MAP therapy for 6 to 24 months. Two trials that used corticosteroids in combination with antimicrobial therapy yielded a pooled odds ratio (OR) of 3.37 for maintenance of remission in treatment versus control, which was statistically significant (95% CI: 1.38–8.24; *p* = 0.013). The subgroup analysis of the other four trials, which did not use corticosteroids to induce remission, yielded a pooled odds ratio of 0.69 (95% CI: 0.39–1.21) for maintenance of remission in treatment versus control, which was not statistically significant (*p* = 0.25). The pooled OR for maintenance of remission in treatment versus control for all six studies was 1.10 (95% CI: 0.69–1.74) in favor of treatment, which was not statistically significant (*p* = 0.78). These results suggest that antimicrobial therapy is effective in maintaining remission in patients with CD after a course of corticosteroids combined with anti-MAP therapy.

Feller et al. [34], in a systematic review and meta-analysis of placebo-controlled trials, examined 13 treatment regimens in 865 patients. The average duration of treatment was 6 months. The outcomes were remission in patients with active disease and relapse in patients with inactive disease. The trials using nitroimidazoles showed benefits, with an OR of 3.54, and the OR for the four trials using clofazimine was 2.86. On the contrary, no benefit was found for classic drugs against tuberculosis (OR = 0.58). The results for clarithromycin were mixed (*p* = 0.005), and in three trials with rifaximin the OR was 2.07. The conclusion of this study was that long-term treatment with nitroimidazoles, clofazimine, or ciprofloxacin appeared to be effective in patients with active CD, while little evidence of benefits was found for clarithromycin and the classical tuberculosis drugs.

Khan et al. [35], in a systematic review including 10 RCTs and 1160 patients, evaluated the effect of antibiotics on remission and relapse of adult patients with active CD. Different kinds of antibiotics were tested, including macrolides, fluoroquinolones, 5-nitroimidazole, and rifaximin, either alone or in combination, for 4 to 16 weeks. There was a statistically significant effect of antibiotics on inducing remission in patients with active CD compared with placebo (OR = 0.85; 95% CI: 0.73–0.99).

Selby et al. [38], in a double-blind, placebo-controlled trial, studied 213 patients with active CD randomized to a 2-year course of daily clarithromycin, rifabutin, and clofazimine or placebo in addition to a 16-week course of prednisolone. The primary endpoint was at least one relapse by 12, 24, or 36 months. Of 122 patients who entered the maintenance phase, 39% who took antibiotics experienced at least one relapse between weeks 16 and 52, compared with 56% who took a placebo (OR = 2.04; *p* = 0.054). The differences between antibiotics and placebo were not statistically significant. The authors concluded that the study did not support a significant pathogenic role for MAP in most CD patients.

The Graham multicenter MAP US study [39] was the first global randomized trial to assess the efficacy of anti-MAP therapy (RHB-104) for 12 months in active CD patients. The anti-MAP therapy, in addition to standard therapy, demonstrated a clinically meaningful and statistically significant treatment effect in the protocol, in which the primary endpoint was defined as remission (CDAI < 150) at week 26, and the secondary endpoint was early remission at week 16 and durable remission through week 52. The remission rate with or without anti-TNF therapy at 26 weeks was significantly higher than placebo (37% vs. 23%, *p* = 0.07). At week 16, the remission rate was 42% vs. 29% (*p* = 0.015).

Agrawal et al. [40], studying a small cohort of pediatric CD patients, concluded that anti-MAP therapy may be more effective than the currently utilized therapies for inducing clinical and endoscopic remission. Although only 47% of patients achieved clinical remission by their first clinical follow-up, 93% of patients achieved remission by the subsequent follow-up appointments after an average of 5 months of treatment (*p* < 0.001).

Lastly, several case series have also been published concerning long-term antibiotic treatment targeting MAP [41,42]. In the Agrawal case series, CD patients experienced profound remission and required no further treatment for 3–23 years [41]. However, the trials and case series produced conflicting results, and no definitive conclusions could be drawn about the favorable effect of anti-MAP therapy on putative MAP infections in CD patients. Moreover, prophylactic antitubercular therapy was found to accelerate disease progression in patients with CD receiving anti-TNF-α therapy [43].

## 4. Antibiotic Treatment Targeting AIEC in Patients with Active CD

Most infections due to intracellular bacteria respond poorly to antibiotic treatment [44]. The lack of antibacterial activity is due to inactivation by the low pH of the phagolysosomes in which antimicrobial bacteria live [45]. Like *Coxiella burnetii*, *Tropheryma whipplei*, and several other bacteria, AIEC also replicates into macrophage phagolysosomes. 

Wiseman et al. [46] first described the effect of pH on the inhibitory activity of chloroquine against *E. coli*. Recently, hydroxychloroquine (HCQ) was found to enhance antibiotic efficacy and macrophage killing of AIEC due to its alkalizing effect on the pH of phagolysosomes [47]. In a study by Flanagan [48], HCQ showed synergistic effects with doxycycline and ciprofloxacin, which are effective antibiotics against intracellular AIEC. Moreover, both HCQ and vitamin D caused dose-dependent inhibition of intramacrophagic AIEC replication 3 h after infection [48].

Rodhes et al. [49], in a randomized trial investigating the treatment of patients with active CD, evaluated prolonged antibiotic treatment with ciprofloxacin, doxycycline, and HCQ for 4 weeks followed by 20 weeks of doxycycline and HCQ, and compared antibiotics with budesonide treatment. The results, including crossover results, showed remission in 9 out of 24 patients treated with HCQ/antibiotics versus only 1 out of 32 patients treated with budesonide. Overall, the results on the efficacy of antibiotic treatment for AIEC-positive CD patients are still scarce and unimpressive. Further clinical trials will be necessary to assess the efficacy of combinations of antibiotics targeting AIEC.

## 5. Short-Term Antibiotic Treatment

Several RCTs utilizing short-term antibiotic treatment for induction and remission of CD produced conflicting results. Steinart et al. [50], analyzing RCTs including 134 patients treated with metronidazole and ciprofloxacin in combination with budesonide, found no differences in remission rates compared with placebo (OR = 1.02; CI: 0.62–1.66) (Table 3). Rahimi et al. [51], in a meta-analysis of broad-spectrum antibiotics, found that patients who received antibacterial therapy for 2 to 24 weeks were 2.257 times more likely to have clinical improvement than those who received placebo (Table 3). Six randomized placebo-controlled trials were included in the meta-analysis. Pulling the results from these trials yielded an OR of 2.157 (CI: 1.678–3.036) for antimicrobial therapy compared with placebo. The conclusion from this study was that broad-spectrum antibiotics improved clinical outcomes in patients with CD.

Prantera et al. [52] studied 402 CD patients after 12 weeks of rifaximin treatment in a clinical trial. After the treatment, 62% of the patients were in clinical remission (*p* < 0.005) (Table 3). Wang et al. [53], in a meta-analysis of broad-spectrum antibiotic therapy, noted clinical improvement in 56% of patients in the antibiotic group and 37.9% in the placebo group after 2–16 weeks of treatment (OR = 1.35 for clinical improvement) (Table 3). Su et al. [54], in a systematic review and meta-analysis, examined 1407 CD patients who received antibiotics for at least 4 weeks, including ciprofloxacin, clarithromycin, metronidazole, and rifaximin. Pooled analysis revealed that, compared with the placebo group, CD patients benefited to a certain extent (RR = 1.32; *p* < 0.00001). However, subgroup analysis showed that there was no significant difference between ciprofloxacin and control (Table 3). Townsend et al. [37] analyzed 13 eligible RCTs comparing antibiotics with a placebo or an active comparator in adult CD patients. Ciprofloxacin, rifaximin, metronidazole, clarithromycin, and cotrimoxazole, alone or in combination, provided only a modest benefit for the induction and maintenance of remission (OD ratio = 0.86 at 6–10 weeks and 0.77 at 10–14 weeks) (Table 3).

Due to the relatively low number of high-quality studies on antibiotics and the high variability in the tested antibiotics, treatment course, and outcome measures, drawing firm conclusions remains difficult.

## 6. Other Therapeutic Strategies Targeting AIEC

Since antimicrobial resistance was observed to affect antibiotics considered to be effective against intracellular AIEC, other possible strategies targeting AIEC have also been proposed:-Anti-adhesive molecules

Monovalent mannosides are promising candidates for use in an alternative and complementary approach for CD patients colonized by AIEC [55]. Type-1 pili are utilized by Gram-negative bacteria to adhere to the host tissue and thus are a key virulence factor in CD. The type-1 pilus was found to mediate the recognition and attachment of AIEC strain to the host [56]. A mannoside recognizing Fim H adhesion, blocking the adhesion of bacteria to cells, was found in the type-1 pilus. A large panel of mannoside-derived Fim H antagonists has been tested to assess the ability of the antagonists to inhibit *E. coli* adhesion to host cells [57].

-Fecal microbiota transplantation

Fecal microbiota transplantation (FMT) is an emerging approach for IBD treatment to restore essential components of the intestinal flora. Modifying the microbial environment by FMT offers an alternative approach that could indirectly influence the host’s immune system in a safe way. One of the newest and least explored methods of modifying the GI microbiota in IBD involves FMT. In the last decade, FMT has undergone a promising transformation, from being considered an alternative form of treatment lacking sufficient medical evidence to be held in reserve, to being accepted as a primary effective therapeutic option.

The FMT procedure involves transferring processed feces from a donor into the gastrointestinal tract of a patient. A recent systematic review and meta-analysis investigated 596 pediatric and adult IBD patients who were enrolled to receive FMT therapy [58]. The pooled estimated clinical remission for CD patients was 30% (CI: 11–52%).

Recently, the efficacy of FMT has been demonstrated in CD patients in independent studies [59,60,61,62]. In a systematic review and meta-analysis, Cheng et al. [63] evaluated the efficacy and safety of FMT treatment in CD patients. Twelve trials were analyzed: after FMT treatment, 0.62% of patients (CI: 0.48–0.51) achieved clinical remission and 0.79% (CI: 0.71–0.89) demonstrated a clinical response. Other adverse events were minor and resolved on their own.

-Probiotics, prebiotics, and postbiotics

The administration of probiotics with presumed anti-inflammatory activity has been tested in CD patients [64], and the efficacy and safety of probiotics for the induction and remission of CD have been reported. As reported in the Cochrane Database of Systematic Reviews [65], after 6 months of treatment there were no significative differences between probiotic treatment and placebo for the induction of remission in CD (OR = 1.06; CI: 0.65–1.71).

Colicin, a species-specific antibiotic, was also investigated. Colicin enters AIEC-containing vacuoles within macrophages and can be delivered either as a purified protein or through colic-producing bacteria. The use of *E. coli* Nissle 1917 as a colicin-producing prebiotic allowed the bacteria to secrete the selected colicin, which is toxic to the AIEC strain [66]. Colicin could potentially be useful to target specific pathogens such as AIEC, where maintaining a healthy microbiome is desirable.

-Phage therapy

Phage therapy is a biological treatment against bacterial infection; however, it targets only a limited number of bacterial strains. An interesting study showed that LF82-P2, LF82-P6, and LF82-P8 phages were effective against AIEC in a mouse model [67]. Galtier et al. [68] found that a single day of oral treatment with bacteriophages significantly decreased intestinal colonization by AIEC strain LF82. Phage therapy has been explored as a promising tool for the eradication of AIEC in CD [69]. Moreover, phage therapy against AIEC in CD patients was found to be safe and effective [70].

-Stem cells

Nowadays, stem cell therapy is widely used to treat CD. Although mesenchymal- and adipose-derived stem cells have proven to be safe for treating CD, there is still a lack of evidence on the efficacy of stem cell therapy for active CD. Moreover, there are still debates on the optimal protocol to use for such therapy in these patients. [71]. Recently, the mechanism of healing of CD patients after mesenchymal stem cell therapy has been reported. 

## 7. Discussion

Based on the effectiveness of antibiotics as well as their favorable adverse effect profile and lower cost compared with biologic drugs or immunomodulators, they provide a more attractive therapeutic option for the treatment of moderate or severe active CD. Generally, traditional antibiotics have shown poor efficacy in active CD, so they are mostly indicated for treating septic complications in the postoperative setting. The rationale for using antibiotics as the primary treatment for CD is based on the increased evidence implicating gut bacteria in the pathogenesis of the disease. However, since the target organism and site of action (intracellular or extracellular) are unknown, the choice of antibiotics can only be arbitrary, and the use of a single antibiotic for short-term treatment can result in antibiotic resistance [44].

Overall, according to the *Antimicrobial consumption in the EU/EEA (ESAC-Net) Annual Epidemiological Report for 2021* [72], in the European Union, *E. coli* was the most common bacterial species (39.4%), with antimicrobial resistance in all reported cases. Antimicrobial agents such as penicillins, cephalosporins, and aminoglycosides, which penetrate poorly into macrophages, are generally ineffective against diseases induced by pathogens that are present within macrophages (Figure 1). On the contrary, azithromycin, ciprofloxacin, clarithromycin, rifampin, sulfamethoxazole, tetracycline, and trimethoprim have been shown to be effective against pathogens such as *E. coli* and MAP internalized by macrophages (Figure 1).

For these reasons, combination therapy using antibiotics that penetrate macrophages may provide a more effective treatment when targeting AIEC [73]. It has been reported that the acid condition of phagolysosomes, in which *E. coli* is located, inhibits antibiotic activity. HCQ, an alkalinizing agent, demonstrated synergistic effects with doxycycline and ciprofloxacin, enhancing the antibiotic efficacy against intramacrophagic AIEC [47,48]. Rodhes et al. [49] found no significant differences in remission or response rates between the antibiotic/HCQ combination and a standard 12-week course of budesonide at 10, 24, or 52 weeks when assessed by intention-to-treat analysis. In that study, to eradicate AIEC in CD patients, ciprofloxacin was used only for 4 weeks and doxycycline was used alone for 20 weeks, which is too short a time to obtain a favorable response. It is our opinion that the unfavorable results of Rhodes’s trial were due to antibiotic resistance. 

Dogan et al. [74] showed that AIEC resistance to one or more antimicrobial agents was present in 75% of CD patients colonized with AIEC and 60% of patients with normal ileum colonized with AIEC (*p* < 0.05). None of the strains were simultaneously resistant to ciprofloxacin, tetracycline, and trimethoprim. AIEC resistance to ciprofloxacin, tetracycline, clarithromycin, rifampicin, and trimethoprim–sulfamethoxazole was found in 25%, 50%, 37.5%, 37.5%, and 50% of CD patients colonized with AIEC, respectively [73]. 

According to a review by Ledder and Turner, the use of ciprofloxacin with or without metronidazole in perianal CD could be valuable as an adjunct to biologics; once again, metronidazole offered benefit in preventing postoperative recurrence in CD patients [75].

It has also been supposed for years that there may be an association between MAP and CD. Several RCTs showed favorable but conflicting results regarding the clinical remission of CD patients after prolonged therapy with multiple anti-MAP drugs [39,40,41]. Unfortunately, in a few trials, MAP detection was performed before treatment, often using inconsistent methods such as culture techniques, which have many limitations, including poor sensitivity. Moreover, in all trials, the primary endpoint of antibiotic treatment was always clinical and endoscopic remission or relapse, evaluated by CDAI and SES-CD.

## 8. Conclusions

In light of the data in the literature, we cannot exclude the notion that the etiopathogenesis of some CD patients may be due to AIEC in some cases and MAP in others, and that the choice of antibiotic treatment for patients with active CD needs to consider the target pathogens. In patients with active CD colonized by AIEC or MAP, a combination of antibiotics that penetrate macrophages should be administered for at least 6 months to avoid antimicrobial resistance. The primary treatment endpoint should be the eradication of pathogens. The secondary endpoint could be clinical and endoscopic remission according to CDAI and SES-CD.

## 9. Future Directions

For all patients with a new diagnosis of CD based on clinical and endoscopic findings, we recommend the detection of AIEC and MAP in ileal/colonic mucosal biopsies using RT-PCR. In patients with active CD and associated AIEC, antibiotic therapy could be administered as a combination of multiple macrophage-penetrating antibiotics. To avoid antibiotic resistance, HCQ could also be used in combination with ciprofloxacin, tetracycline, and trimethoprim for at least 6 months (Figure 2).

For patients with active CD and associated MAP, we suggest long-term (up to 6 months) anti-MAP treatment with rifabutin, clarithromycin, and clofazimine (Figure 2). For all CD patients colonized with AIEC or MAP treated with antimicrobial therapy, the primary treatment endpoint should be the eradication of AIEC or MAP, as assessed by RT-PCR (Figure 2). The secondary endpoint should be clinical and endoscopic remission, as evaluated by CDAI and SES CD.

Finally, conventional therapy could be suggested only for CD patients without associated AIEC or MAP (Figure 2).

## Figures and Tables

**Figure 1 antibiotics-13-00151-f001:**
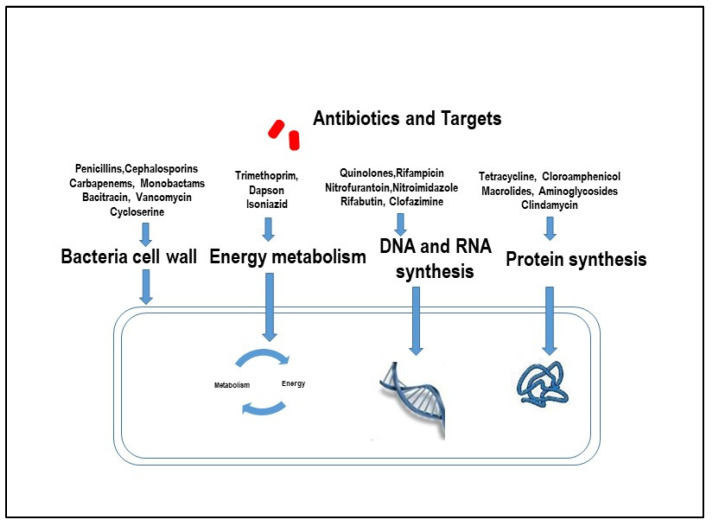
Mechanisms by which antibiotics act on harmful pathogens.

**Figure 2 antibiotics-13-00151-f002:**
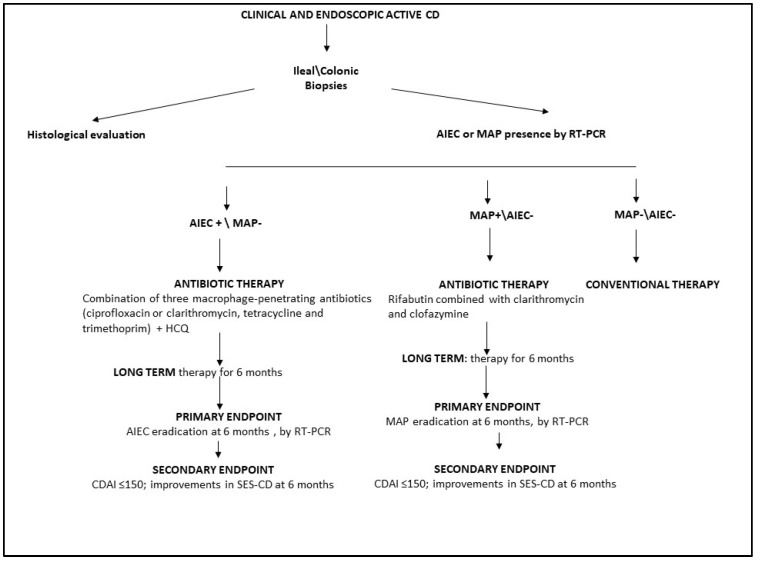
Schematic workflow for antibiotic treatment of patients with active CD. HCQ, hydroxychloroquine; AIEC, adherent invasive *E. coli*; MAP, *Mycobacterium avium paratuberculosis*; CDAI, Crohn’s Disease Activity Index; SES-CD, Simple Endoscopic Score for Crohn’s Disease; RT-PCR, real-time polymerase chain reaction.

**Table 1 antibiotics-13-00151-t001:** Pathogens potentially involved in CD.

Bacteria	References
✓ *Yersinia enterocolitica*	[2]
✓ *Helicobacter species*	[2]
✓ *Campylobacter* species	[2]
✓ *Listeria monocytogenes*	[2]
✓ *E. coli* species	[8,9,10,11,12,13]
✓ *Mycobacterium avium paratubercolosis*	[14,15]

**Table 2 antibiotics-13-00151-t002:** Long-term antibiotic treatment targeting MAP in patients with active CD.

Author	Number of Trials	Number of Patients	Antibiotics	Duration	Placebo or Other Comparators	Primary Outcome	OR
Borgoankar [33]	6	317	Anti-MAP + corticosteroids (2 trials)	6–24 months	-	CDAI < 150	1.10 (0.69–1.74) (all trials)
		865	Anti-MAP + standard therapy (4 trials)				3.37 (1.38–8.24) (2 trials)
Feller [34]	16	58	Rifaximin (1 trial)	3 months	Placebo	CDAI < 150	2.07 (0.71–6.06)
		206	Nitroimidazole (3 trials)	3–24 months	Placebo	CDAI < 150	3.54 (1.94–6.47)
		322	Clofazimine (4 trials)	3–24 months	Placebo	CDAI > 70 from baseline	2.86 (1.67–4.88)
		287	Clarithromycin alone or in combination (4 trials)	3–24 months	Placebo	CDAI < 150	0.58 (0.29–1.18)
		107	Anti-tuberculosis drugs (3 trials)	3–24 months	Placebo	CDAI < 150	11.3 (2.60–48.8)
		47	Ciprofloxacin (1 trial)	6 months	Placebo	CDAI < 150	0.85 (0.73–0.99)
Khan [35]	10	1160	Macrolides, fluorochinolones, 5-nitromidazole, Rifaximin alone or in combination	1–4 months	Placebo	CDAI < 150	0.85
Selby [38]	1	213	Rifabutin, clarithromycin, and clofazimine (AMAT)	16–104 months	Placebo + 16 weeks tapering course Prednisolone	At least 1 relapse between 16 and 52 weeks	2.04 (0.84–4.93)
Graham [39]	1	331	RHB104: rifabutin, clarithromycin, or Clofazimine + anti-TNF or azatioprine or 6-mercaptopurine + 5 ASA corcorticosteroids (tapering after 8 weeks)	12 months	Placebo	CDAI < 150	at 26 weeks
Agrawal [40]	1	16	Rifabutin, clarithromycin, clofazimine + metronidazole or ciprofloxacin	5 months	-	wPCDAI: 47.5	-

OR, odds ratio; CDAI, Crohn’s Disease Activity Index; and wPCDAI, Weighted Pediatric Crohn’s Disease Activity Index.

**Table 3 antibiotics-13-00151-t003:** Short-term antibiotic treatment for patients with active CD.

Author	Number of Trials	Number of Patients	Antibiotics	Duration	Placebo or Other Comparators	Primary Outcome	OR
Steinhart [50]	1	134	Metronidazole, ciprofloxacin, budesonide	8 weeks	Placebo	CDAI < 150	-
Rahimi [51]	6	804	Metronidazole, ciprofloxacin, Cotrimoxazole alone (2 trials) or in combination (4 trials)	2–24 weeks	Placebo	CDAI < 150	2.257
Prantera [52]	1	402	Rifaximin	12 weeks	Placebo	CDAI < 150	-
Wang [53]	-	83	Ciprofloxacin, metronuidazole alone or in combination, rifaximin, clarithromycin	2–16 weeks	Placebo	CDAI < 150	1.35
Su [54]	15	1407	Ciprofloxacin, fluoroquinolones, clarithromycin, metronidazole, rifaximin	at least 4 weeks	Placebo	CDAI < 15	1.35
Townsend [37]	13	1303	Rifaximin, clarithomycin, metronidazole, cotrimoxazole, Anti-MAP alone or in combination with budesonide	6–14 weeks	Placebo alone or in combination	CDAI < 150	0.77 to 0.33

OR, odds ratio; CDAI, Crohn’s Disease Activity Index.

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
