# Peer review of "Antibiotic Therapy for Active Crohn’s Disease Targeting Pathogens: An Overview and Update"

_antibiotics, 2024, doi:10.3390/antibiotics13020151_

Round 1
Reviewer 1 Report
Comments and Suggestions for Authors
A decent review.
The following are suggested to enhance the manuscript:
1. FMT being part of antibiotics in IBD is unusual. Please clarify
2. The following may be referenced and discussed:
a. Ledder O, Turner D. Antibiotics in IBD: Still a Role in the Biological Era? Inflamm Bowel Dis. 2018 Jul 12;24(8):1676-1688. doi: 10.1093/ibd/izy067. PMID: 29722812.
b. Khan KJ, Ullman TA, Ford AC, Abreu MT, Abadir A, Marshall JK, Talley NJ, Moayyedi P. Antibiotic therapy in inflammatory bowel disease: a systematic review and meta-analysis. Am J Gastroenterol. 2011 Apr;106(4):661-73. doi: 10.1038/ajg.2011.72. Epub 2011 Mar 15. Erratum in: Am J Gastroenterol. 2011 May;106(5):1014. Abadir, A [corrected to Abadir, Amir]. PMID: 21407187.
Comments on the Quality of English LanguageEnglish is largely acceptable
Author Response
Reviewer 1 A decent review.
The following are suggested to enhance the manuscript:
- FMT being part of antibiotics in IBD is unusual. Please clarify
- Reply
- FMT was not included among antibiotics, but in paragraph 6 Other therapeutic strategies targeting AIEC, together with stem cells, phage therapy, etc..
- 2. The following may be referenced and discussed: a. Ledder O, Turner D. Antibiotics in IBD: Still a Role in the Biological Era? Inflamm Bowel Dis. 2018 Jul 12;24(8):1676-1688. doi: 10.1093/ibd/izy067. PMID: 29722812. b. Khan KJ, Ullman TA, Ford AC, Abreu MT, Abadir A, Marshall JK, Talley NJ, Moayyedi P. Antibiotic therapy in inflammatory bowel disease: a systematic review and meta-analysis. Am J Gastroenterol. 2011 Apr;106(4):661-73. doi: 10.1038/ajg.2011.72. Epub 2011 Mar 15. Erratum in: Am J Gastroenterol. 2011 May;106(5):1014. Abadir, A [corrected to Abadir, Amir]. PMID: 21407187.
- Reply
- The first reported article was discussed and introduced among the references (75), while we had already talked about the second work in paragraph 3, referencing it as n. 35.

Reviewer 2 Report
Comments and Suggestions for Authors
The article is interesting, it summarizes knowledge on the subject antibiotic therapy for active Crohn’s disease targeting pathogens.
Needs to be corrected:
71 - E.Coli species - change to: E.coli species
74- of E.Coli- change to: of E.coli
Table 1. -Escherichia Coli species - change to: E.coli species
Author Response
Reviewer 2 The article is interesting, it summarizes knowledge on the subject antibiotic therapy for active Crohn’s disease targeting pathogens. Needs to be corrected: 71 - E.Coli species - change to: E.coli species 74- of E.Coli- change to: of E.coli Table 1. -Escherichia Coli species - change to: E.coli species
Reply
We thank the reviewer. All typos have been corrected.

Reviewer 3 Report
Comments and Suggestions for Authors
Dear,
this is an interesting topic, and hope will catch the eye of the wider audience. However, can you provide us some real world data from clinical practice and particular use of this agents. How many hospitals provide this modality of care for CD patients? What about adverse effects of AB in this patients, such as colitis, etc.
Comments on the Quality of English LanguageSentence should be organized to be more precise, particular when you describe the results of the trials.
Author Response
Reviewer 3
Dear,
this is an interesting topic, and hope will catch the eye of the wider audience. However, can you provide us some real world data from clinical practice and particular use of this agents. How many hospitals provide this modality of care for CD patients? What about adverse effects of AB in this patients, such as colitis, etc.
Reply
To date, there are no hospitals that provide antibiotic therapy for patients with active Crohn's disease, because it is not recommended by International guidelines.
Recently, European Crohn’s and Colitis Organisation (ECCO) guidelines on therapeutics in Crohn’s disease (Torres J, Bonovas S, Doherty G, et al. ECCO Guidelines on Therapeutics in Crohn's Disease: Medical Treatment. J Crohns Colitis. 2020;14(1):4--22. doi:10.1093/ecco--jcc/jjz180) suggest against the use of antibiotics in active Crohn’s disease, because no studies demonstrated efficacy to consistently induce clinical remission or mucosal healing compared to placebo.
When antibiotic therapy was used in clinical trials or case series for patients with active CD, the treatment was well tolerated , but adverse events such as colitis, abnormal liver function, vaginal candidosis, myalgia, arthralgia, etc..., were significantly more common in the antibiotic treated patients than in the placebo patients.
Comments on the Quality of English Language
Sentence should be organized to be more precise, particular when you describe the results of the trials.
Reply
The manuscript was edited by MDPI's English pre--edit services, as shown in the attached certificate of editing.
